# Influence of Cannabinoid Receptor 1 Genetic Variants on the Subjective Effects of Smoked Cannabis

**DOI:** 10.3390/ijms22147388

**Published:** 2021-07-09

**Authors:** Thomas Murphy, Justin Matheson, Robert E. Mann, Bruna Brands, Christine M. Wickens, Arun K. Tiwari, Clement C. Zai, James Kennedy, Bernard Le Foll

**Affiliations:** 1Centre for Addiction and Mental Health, Translational Addiction Research Laboratory, University of Toronto, 33 Ursula Franklin Street, Toronto, ON M5S 2S1, Canada; Thomas.Murphy@camh.ca; 2Department of Pharmacology and Toxicology, University of Toronto, Toronto, ON M5S 1A8, Canada; justin.matheson@camh.ca (J.M.); bruna.brands@camh.ca (B.B.); christine.wickens@camh.ca (C.M.W.); 3Centre for Addiction and Mental Health, Institute for Mental Health Policy Research, Toronto, ON M6J 1H4, Canada; robert.mann@camh.ca; 4Controlled Substances and Cannabis Directorate, Health Canada, Ottawa, ON K1A 0K9, Canada; 5Centre for Addiction and Mental Health, Campbell Family Mental Health Research Institute, Toronto, ON M6J 1H4, Canada; 6Dalla Lana School of Public Health, University of Toronto, Toronto, ON M5T 3M7, Canada; 7Institute of Health Policy, Management and Evaluation, University of Toronto, Toronto, ON M5T 3M6, Canada; 8Neurogenetics Section, Tanenbaum Centre for Pharmacogenetics, Molecular Brain Science, Campbell Family Mental Health Research Institute, CAMH, Toronto, ON M5T 1R8, Canada; arun.tiwari@camh.ca (A.K.T.); clement.zai@camh.ca (C.C.Z.); jim.kennedy@camh.ca (J.K.); 9Department of Psychiatry, University of Toronto, Toronto, ON M5T 1R8, Canada; 10Institute of Medical Science, University of Toronto, Toronto, ON M5S 1A8, Canada; 11Laboratory Medicine and Pathobiology, University of Toronto, Toronto, ON M5S 1A8, Canada; 12T.H. Chan School of Public Health, Harvard University, Boston, MA 02115, USA; 13Broad Institute, Cambridge, MA 02142, USA; 14Acute Care Program, Centre for Addiction and Mental Health, Toronto, ON M6J 1H4, Canada; 15Department of Family and Community Medicine, University of Toronto, Toronto, ON M5G 1V7, Canada

**Keywords:** cannabis, Δ^9^-tetrahydrocannabinol, subjective effects, *CNR1* gene

## Abstract

As many jurisdictions consider relaxing cannabis legislation and usage is increasing in North America and other parts of the world, there is a need to explore the possible genetic differences underlying the subjective effects of cannabis. This pilot study investigated specific genetic variations within the cannabinoid receptor 1 (*CNR1*) gene for association with the subjective effects of smoked cannabis. Data were obtained from a double-blinded, placebo-controlled clinical trial studying the impact of cannabis intoxication on driving performance. Participants randomized to the active cannabis group who consented to secondary genetic analysis (*n* = 52) were genotyped at the *CNR1* rs1049353 and rs2023239 polymorphic areas. Maximum value and area under the curve (AUC) analyses were performed on subjective measures data. Analysis of subjective effects by genotype uncovered a global trend towards greater subjective effects for rs1049353 T-allele- and rs2023239 C-allele-carrying subjects. However, significant differences attributed to allelic identity were only documented for a subset of subjective effects. Our findings suggest that rs1049353 and rs2023239 minor allele carriers experience augmented subjective effects during acute cannabis intoxication.

## 1. Introduction

Cannabis has been used medically for centuries, yet today cannabis is used primarily for recreational purposes [1]. For much of the 20th century, cannabis was illegal in most countries of the world, yet it is now the most widely used drug after alcohol and tobacco [2]. In the United States, peak usage occurred in the late 1970s and the following decades exhibited a decline in cannabis use due to increased education surrounding risks and widespread public disapproval [3,4]. Nonetheless, since 1992, there has been a resurgence of cannabis usage in Canada and the United States. During prohibition, it was reported that underage Canadians had greater access to cannabis than alcohol and were nearly twice as likely to experiment with cannabis over tobacco, and Canadian adolescents had the highest rates of cannabis consumption in the industrialized world [5,6]. Youth cannabis consumption in Canada was relatively stable between 2004 and 2015. However, consumption amongst the 25–44-year-old age group increased, from approximately 10% to 18% of the population, a trend in adult consumption that has been previously documented in Canada and the US [7,8,9]. In 2017, cannabis was the most widely used illicit substance worldwide and was used by an estimated 188 million, or 3.8% of the global adult population, an estimated 30% increase from two decades prior [10]. North America exhibited the largest increases in rates of cannabis use, with usage rates rising from 7% to 8.4% between 2007 and 2017. Cannabis usage rates have remained relatively stable in other parts of the world: ~10% in Oceania, ~7% in Europe, and ~2% in Asia.

Throughout North America, there has been a progressive relaxation of cannabis legislation during the last two decades. In the US, at the time this manuscript was written, recreational cannabis use has been legalized in nearly a dozen states and decriminalized in many more. In 2001, medical use of cannabis was made available in Canada; in October 2018, recreational use of cannabis was legalized, with edibles, oils, and topicals legalized a year later. Canada’s *National Cannabis Survey*, *first quarter 2019*, reported that approximately 18% of Canadians aged 15 years and older (5.3 million) had used cannabis in the first quarter after legalization, higher than the 14% reported in the first quarter of 2018 [11]. Interestingly, much of this increase is attributable to greater use in individuals aged 45 to 64 years, many of whom were first-time users. Additionally, males exhibited an increase in cannabis use during this time, from 16% to 22%, while usage in females remained stable at 13% [12]. While smoking of dried cannabis leaves remains the most common form of consumption in North America, legalization of cannabis has also increased prevalence of other consumption methods, including edible product consumption and vaporization [13,14].

The endocannabinoid system (ECS) is a homeostatic biological system that controls the activity of various excitatory and inhibitory neurotransmitter systems including the glutamatergic, serotoninergic, dopaminergic, noradrenalin, acetylcholine, and gamma-aminobutyric acid (GABA) systems [2,15]. The ECS encompasses two main G protein-coupled receptor (GPCR) subtypes, CB1 and CB2, the endocannabinoid lipid-based ligands, 2-arachidonoylglycerol (2-AG) and anandamide (AEA), and all enzymes and biochemical pathways involved in their biosynthesis and metabolism [16,17]. The CB1 receptor is 472 amino acids in length, is encoded by the cannabinoid receptor 1 (*CNR1*) gene, and is highly expressed in the brain and central nervous system, yet also detectable at lower concentrations in areas such as the liver and pancreas [18,19]. The CB1 receptor of the ECS, while a receptor of the endocannabinoids, 2-AG and AEA, is also responsible for mediating the effects of cannabis. Δ^9^-Tetrahydrocannabinol (THC), the main psychoactive component of cannabis is a partial agonist of the CB1 receptor (K_i_ = 27.1 nM) [20]. THC agonism at the CB1 receptor induces many of the subjective effects of cannabis including sedation, analgesia, antiemesis, anxiolysis, appetite stimulation, and psychotropic effects [21,22].

Genetic variation of the *CNR1* gene may generate populational variations in the structure and function of the resultant receptor, thus eliciting alterations in CB1 signaling and the effects of cannabis. A single-nucleotide polymorphism (SNP) is a genetic variation occurring at a single nucleotide, whereby the genetic base differs from the typical base exhibited in other members of the same species. In recent decades, numerous SNPs have been identified within the *CNR1* gene, and their effects on drug consumption patterns and effects have been studied.

The rs1049353 polymorphism is a well-characterized C/T substitution at the 88143916 position of chromosome 6. Longitudinal assessment of an adolescent population (*n* = 88) over 13 years discovered that the presence of the minor T-allele significantly increased the level of cannabis use, and that multiple T-alleles additively elevated cannabis usage [23]. One recent investigation discovered that rs1049353 T-allele carriers exhibit significantly lower state satiety following THC exposure compared to CC individuals, possibly signifying augmented subjective effects, resulting in craving for additional cannabis [24]. Conversely, sociodemographic data analysis has suggested that the CC genotype is more prominent in cannabis users and the C-allele confers greater risk of becoming a cannabis user [25]. The Profile of Mood States (POMS) scale has been used to assess subjective effects on mood pre- and post-cannabis smoking and it was discovered that C carriers at the rs1049353 polymorphism experience elevated *Anger*/*Hostility* following cannabis exposure [26]. The major C-allele of the rs1049353 polymorphism has been associated with cannabis dependence symptoms. However, this relationship did not survive correction, and neither allele has been associated with cannabis use disorder [27,28].

The rs2023239 T/C substitution is another comprehensively studied variant, located within the *CNR1* gene at the 88150763 position of chromosome 6. The rs2023239 polymorphism has been documented as affecting the expression and conformation of the CB1 receptor, with a greater implication of having a modulatory effect over endocannabinoid signaling. Positron Emission Tomography (PET) study of CB1 receptor binding discovered that minor C-allele carriers exhibit elevated CB1 receptor binding compared to major allele TT individuals [29]. This finding is corroborated by Hutchison et al., who studied postmortem brain tissues and reported enhanced C carrier CB1 binding in the prefrontal cortex of the brain, which they hypothesized may be due to elevated CB1 receptor expression [30]. This hypothesis was recently confirmed, as a lymphocyte-based assay to evaluate CB1 concentration determined minor C-allele-carrying cannabis users possess greater CB1 receptor concentration compared to TT users and nonusers [31]. The aforementioned investigation that utilized the POMS scale to study its modulation following cannabis consumption similarly genotyped the rs2023239 polymorphic site and analyzed results by allelic identity. The findings indicate that minor C-allele carriers exhibit elevated *Anger*/*Hostility*, *Fatigue*, *Tension*/*Anxiety*, and *Vigor* scores over TT individuals [26]. Regular cannabis users exposed to cannabis-related cues during a period of abstinence have been monitored by fMRI, demonstrating genotypic modulation of brain activity, measured by a blood oxygenated level-dependent (BOLD) response. Interestingly, in response to these cues, C-carrying subjects display significantly greater BOLD response over TT subjects in reward-related brain regions, including the orbitofrontal cortex (OFC), inferior frontal gyrus (IFG), insula, and dorsal anterior cingulate gyrus (ACG) [32]. Additionally, rs2023239 C-allele-carrying cannabis users experience greater withdrawal after short-term abstinence, higher craving after smoking and cannabis cue exposure, and greater mean usage [33].

Following a thorough review of the literature pertaining to specific candidate genes and their association with the subjective effects of acute cannabis exposure, it is clear that relatively few studies have pursued this research area. The *CNR1* gene appears to be a logical area to examine in relation to drug use, abuse, and related phenotypes. While the rs1049353 and rs2023239 polymorphisms are amongst the best-characterized *CNR1* polymorphisms, only a handful of studies have focused on them in this context, yielding discrepant results. Considering the present climate of relaxing cannabis legislation and rising usage amongst the North American population and around the world, there is need to understand the genetic basis underlying the subjective effects of cannabis. Therefore, the goal of the present study was to perform a genetic analysis of the effects of genetic variation in the *CNR1* gene on the subjective effects of smoked cannabis in a sample of young adults. In this study, we genotyped the *CNR1* rs1049353 and rs2023239 variants, and correlated the presence of allelic variants to Visual Analogue Scale (VAS) and the Profile of Mood States (POMS) subjective measure results. This study will serve to replicate results from other studies and add to the body of work surrounding the genetics behind variable effects of cannabis.

We hypothesize that following acute exposure to smoked cannabis, the rs1049353 major C-allele and rs2023239 minor C-allele will be associated with stronger indicators of subjective effects of the drug.

## 2. Results

### 2.1. Subject Pool and Genotyping Results

A total of 178 subjects were assessed for eligibility in the original trial [34,35]. Of the assessed individuals, 99 participants were enrolled and randomized to the active cannabis or placebo group, of which 73 provided consent to participate in the supplemental analysis of genetic influence on subjective effects of cannabis. In that group, 3 were excluded from final analysis due to issues including missing baseline data and substance consumption between sessions, leaving 70 completers (52 active, 18 placebo) that were included in the final analysis. Placebo group participants were excluded, leaving a final sample size of 52 participants. Demographic characteristics of the sample are displayed in Table 1.

Twenty-nine subjects homozygous for the CNR1 rs1049353 CC-allele (19 males and 10 females) were compared to 23 subjects carrying the minor T-allele (17 males and 6 females). Thirty-five subjects homozygous for the CNR1 rs2023239 major T-allele (25 males and 10 females) were compared to 17 subjects carrying the minor C-allele (11 males and 6 females). Subject group composition is summarized in Table 2.

### 2.2. Blood THC Analysis

For rs1049353, the genotype by time interaction was not significant when the Greenhouse–Geisser correction was applied (*p* = 0.083). However, there was a significant main effect of genotype (*p* = 0.039). T-tests indicated that the genotype difference was significant at 15 min (*p* = 0.04), 4 h (*p* = 0.031), 5 h (*p* = 0.039), and 6 h (*p* = 0.022). As displayed in Figure 1A, in all cases, blood THC concentrations were higher in the minor allele group (TT + TC) than in the homozygous CC group. When analyzed the PK parameters, the minor allele group also had a higher AUC for THC than the major allele homozygous group (CC—19.36 ± 3.72, TC + TT—34.06 ± 5.92; *p* = 0.035), but the higher Cmax in the TT + TC group was not statistically significant (CC—33.41 ± 5.65, TC + TT—47.44 ± 7.47; *p* = 0.14). A *t*-test revealed T-carrying individuals consumed a larger quantity of cannabis during the 10 min smoking period than major allele homozygotes (CC—76.23 mg ± 21.86, TC + TT—90.93 ± 16.53; *p* = 0.01).

For rs2023239, as displayed in Figure 1B, there was no significant genotype by time interaction (*p* = 0.92) or main effect of genotype (*p* = 0.79). Similarly, there was no significant difference in THC AUC or Cmax.

### 2.3. Visual Analogue Scales

#### 2.3.1. CNR1 rs1049353 Polymorphism

Figure 2A summarizes Visual Analogue Scale (VAS) mean maximum values assessed following cannabis administration, separating subjects by genotype at the rs1049353 position. It is clear that individuals carrying the rs1049353 minor T-allele scored higher on all seven VAS items than individuals homozygous for the major C-allele. However, following univariate analysis of variance (ANOVA), only one VAS item, *I like the drug* (CC—68.207 ± 4.762, TC + TT—84.727 ± 5.467; *p* = 0.027) demonstrated a significant difference by genotype.

Figure 2B summarizes VAS item time course graph AUC values assessed following cannabis administration, created using mixed-effects model analysis, separating participants by genotype at the rs1049353 position. Using the mixed-effects model generated results, differences in AUC between rs1049353 genotype were calculated. It is clear that individuals carrying the rs1049353 T minor allele scored higher on all seven VAS items than individuals homozygous for the major allele. However, following AUC calculation, the *I like the drug* VAS item ((AUC TC + TT) − (AUC CC) = 4848.81 ± 2251.74; *p* = 0.036) was the only item with a significant difference by genotype. Figure 2C displays the mixed-effects model time course graph for the *I like the drug* item.

#### 2.3.2. CNR1 rs2023239 Polymorphism

Figure 3A summarizes VAS maximum values assessed during acute cannabis intoxication, separating subjects by genotype at the rs2023239 position. It is apparent the rs2023239 C minor allele-carrying individuals scored higher on all seven VAS items than major allele homozygous TT individuals. However, following univariate ANOVA analysis, none of the VAS items showed significant differences by genotype.

Figure 3B summarizes VAS item time course graphs assessed during acute cannabis intoxication, created using mixed-effects model analysis, separating subjects by genotype at the rs2023239 position. Like previous results have shown, rs2023239 C-allele-carrying individuals appear to have scored higher on all seven VAS items than homozygous TT individuals. However, following AUC calculation, only the *I like the drug* ((AUC CT + CC) − (AUC TT) = 6703.82 ± 2328.62; *p* = 0.006) and *It feels like cannabis* ((AUC CT + CC) − (AUC TT) = 8732.80 ± 2692.45; *p* = 0.002) items exhibited significant differences in AUC by genotype. Figure 3C,D display the mixed-effects model time course graphs for the *I like the drug* and *It feels like cannabis* items.

#### 2.3.3. Adjusted Results

Univariate ANOVA analysis maximum value results for rs1049353 were adjusted by controlling for participant sex, BMI, and estimated THC dose administered. Following adjustment, significance was not retained for the *I like the drug* VAS item (CC—71.831 ± 5.352, TC + TT—82.839 ± 5.864; *p* = 0.170). Mixed-effects model AUC results for rs1049353 were also adjusted by controlling for the same factors. Following adjustment on AUC results, significance was not retained for the *I like the drug* item ((AUC TC + TT) − (AUC CC) = 3207.94 ± 2302.97; *p* = 0.170).

Following adjustment of maximum value results for rs2023239, none of the VAS items showed any significant differences by genotype. Following adjustment of rs2023239 AUC results, significance was retained in both VAS items that displayed unadjusted genotypic group differences: *I like the drug* ((AUC CT + CC) − (AUC TT) = 6046.42 ± 2229.17; *p* = 0.009) and *It feels like cannabis* ((AUC CT + CC) − (AUC TT) = 8730.01 ± 2546.38; *p* = 0.001).

### 2.4. Profile of Mood States

#### 2.4.1. CNR1 rs1049353 Polymorphism

Maximum values of the POMS subscales, as a function of rs1049353 genotype, are displayed in Figure 4. As with the VAS, visual inspection suggests that T-allele heterozygous and homozygous individuals scored higher on all ten POMS subscales than individuals homozygous for the major C-allele. Univariate ANOVA analysis revealed significant genotypic differences in the *Anger*/*Hostility* (CC—0.146 ± 0.066, TC + TT—0.391 ± 0.075; *p* = 0.018), *Depression*/*Dejection* (CC—0.092 ± 0.06, TC + TT—0.315 ± 0.067; *p* = 0.016), *Vigor* (CC—1.198 ± 0.122, TC + TT—1.826 ± 0.138; *p* = 0.001), and *Elation* (CC—1.414 ± 0.126, TC + TT—1.862 ± 0.141; *p* = 0.021) subscales.

Split-plot ANOVA analyses were conducted to analyze the genotypic significance of POMS scores between baseline and 1 h post-cannabis exposure. Following split-plot ANOVA analyses, none of the POMS subscales displayed significant changes between genotypic groups.

#### 2.4.2. CNR1 rs2023239 Polymorphism

The same analysis was performed on maximum values of POMS subscales, separating subjects by rs2023239 genotype, and the results are displayed in Figure 5. Similar to previous results, there is a trend towards higher POMS subscale maximum values in minor allele C carriers. However, following univariate ANOVA analysis, none of the subscales exhibited statistically significant differences by genotype.

Split-plot ANOVA analyses were conducted to analyze the genotypic significance of POMS scores between baseline and 1 h post-cannabis exposure. Following split-plot ANOVA analyses, none of the POMS subscales displayed significant changes between genotypic groups.

#### 2.4.3. Adjusted Results

Univariate ANOVA analysis maximum value results for rs1049353 were adjusted by controlling for participant sex, BMI, and estimated THC dose administered. Following adjustment, significance was not retained for the *Elation* (TT—1.458 ± 0.136, CT + CC—1.748 ± 0.147; *p* = 0.148) item but was retained for the *Vigor* (TT—1.297 ± 0.134, CT + CC—1.787 ± 0.145; *p* = 0.015) and *Depression*/*Dejection* (TT—0.052 ± 0.068, CT + CC—0.309 ± 0.073; *p* = 0.012), and *Anger*/*Hostility* (TT—0.088 ± 0.073, CT + CC—0.370 ± 0.080; *p* = 0.011) POMS items. Split-plot ANOVA results for rs1049353 were also adjusted by controlling for the same factors. Following adjustment, significance was not exhibited for any of the POMS items.

Following adjustment of maximum value results for rs2023239, none of the POMS items displayed any significant differences by genotype. Following adjustment of rs2023239 split-plot ANOVA results, none of the POMS items displayed any significant differences by genotype.

## 3. Materials and Methods

### 3.1. CADRI Study

Blood samples for genetic analysis were collected during the clinical trial entitled, *Acute and residual effects of cannabis on young drivers’ performance of driving related skills* (herein referred to as CADRI). The CADRI study was approved by the CAMH Research Ethics Board and Health Canada Ethics Board, and was a double-blinded, placebo-controlled mixed-design study with the primary objective of studying the acute and residual effects of a moderate dose of smoked cannabis (12.5% THC) and its effects on driving behaviour and performance in a laboratory setting [34,35]. One of the secondary aims of the CADRI trial was to study whether a relationship exists between genetics and the subjective effects of smoked cannabis. Informed consent was obtained prior to commencement of the CADRI eligibility assessment and the informed consent form included a supplemental request for the participant to provide two additional blood samples (~20 mL) for use in future genetic analyses.

The CADRI study population was composed of healthy young adults (males and females) that were aged 19–25, held a valid driver’s license, and were regular cannabis users (1–4 days per week). Potential participants were recruited from community advertisements within the Greater Toronto Area and initially underwent a telephone eligibility screen with a member of the study team and attended CAMH for an eligibility assessment. Participants were excluded if they were regular users of medications that affect brain function (i.e., antidepressants, benzodiazepines, stimulants, etc.), had been diagnosed with severe medical or psychiatric conditions, were pregnant, trying to become pregnant, breastfeeding, or met criteria for lifetime substance dependence (Structured Clinical Interview for DSM-IV Axis I Disorders), including cannabis. Once enrolled, participants were randomized into active and placebo groups in a 2:1 allocation ratio to maximize active group participants. Participants were asked to abstain from cannabis use 48 h prior to, and during the course of the trial.

Active cannabis (12.5% THC) was obtained from Prairie Plant Systems (Saskatoon, SK, Canada) and matching placebo cannabis (<0.1% THC) was obtained from the National Institute on Drug Abuse (NIDA) (North Bethesda, MD, USA). Each cigarette contained 750 mg of plant material. Subjects were escorted to a reverse airflow room that facilitated external ventilation of expired smoke. Subjects smoked alone and were instructed to smoke as they usually do for a maximum of 10 min and were told to stop smoking if they felt ill or achieved a high greater than they would normally experience. At the end of the smoking period, unsmoked cannabis cigarette remnants were collected and weighed. Estimated THC dosage was then calculated by subtracting this weight from 750 mg and multiplying by the 12.5% THC concentration. The acute intoxication period was defined up until 6 h post-administration and subjects returned and underwent assessment at 24 and 48 h post-dose.

During the CADRI study, the following data were collected to measure the subjective effects participants experienced from smoked cannabis:Self-reports of drug effects using Visual Analog Scales (VAS)
○VAS questions: (1) I feel a drug effect, (2) I feel this high, (3) I feel the drug’s good effects, (4) I feel the drug’s bad effects, (5) I like the drug, (6) I feel a rush, and (7) It feels like cannabis.○Assessed on a 10 cm continuum and scored from 0 to 100.○Baseline, 5, 15, 30 min and 1, 2, 3, 4, 5, 6, 24, and 48 h post-dose.Profile of Mood States (POMS) [26,36]
○Psychometric rating scales used to assess a variety of transient mood states.○POMS scales: (1) Tension/Anxiety, (2) Anger/Hostility, (3) Depression/Dejection, (4) Friendliness, (5) Fatigue, (6) Confusion, (7) Vigor, (8) Elation, (9) Arousal, and (10) Positive Mood.○Baseline, 1, 24, and 48 h post-dose.

For those participants that consented during the eligibility assessment to the supplemental study of genetic influences on the effects of cannabis, an additional 20 mL of blood was drawn during the baseline blood draw of the Testing Day 1 session and DNA was extracted.

### 3.2. Genotyping Experiment

Following CAMH Research Ethics Board approval, a genotyping experiment was performed on the DNA samples of consenting participants. The Infinium Global Screening Array (GSA) ChiP microarray from Illumina (San Diego, CA, USA) was used to genotype specific polymorphic sites that have been associated with cannabis use, abuse, and variable subjective response, with a primary focus on the rs1049353 and rs2023239 SNPs of the cannabinoid receptor 1 (*CNR1*) gene. The GSA data went through the following quality control steps: matching between self-reported sex and genotype-predicted sex, SNP-level and subject-level missingness of less than 5%, SNP minor allele frequencies of at least 5%, SNP genotypes not deviating significantly from the Hardy–Weinberg Equilibrium (*p* > 5 × 10^−8^), subject heterozygosity of less than four standard deviations from the mean, checking for related subjects, and ancestry of the subjects.

### 3.3. Statistical Analysis

Following the genotyping experiment and quality control, subjects were grouped by genotypes at the *CNR1* rs1049353 and rs2023239 SNPs. Considering our primary research interest is how these polymorphisms affect the subjective effects of smoked cannabis, placebo group subjects were excluded. With a relatively small sample size of active group participants, we compared two groups for both the rs1049353 and rs2023239 polymorphisms: (1) subjects carrying at least one copy of the minor allele and (2) subjects homozygous for the major allele. Analysis was then conducted between groups based on data on the subjective effects of acute smoked cannabis intoxication.

We base genotype power considerations on allele frequencies of the rs1049353 T-allele and the rs2023239 C-allele. The Allele Frequency Generator (ALFA) on the NCBI database reports that the global frequency for these minor alleles is 25.2% (*n* = 145,948) and 22.1% (*n* = 11,360), respectively [37]. Beginning with the rs1049353 polymorphism, with 52 completers tested, we expect approximately 13 to be T-allele (CT or TT genotype) carriers. With group sizes of 39 and 13 and a significance level of 0.05, we can detect an effect size of 0.915 with 80% power for the rs1049353 polymorphism. For the rs2023239 polymorphism, we expect approximately 11 to be C-allele (TC or CC genotype) carriers. With group sizes of 41 and 11 and a significance level of 0.05, we can detect an effect size of 0.970 with 80% power for the rs2023239 polymorphism.

Statistical analyses were performed using IBM SPSS Statistics (Armonk, NY, USA). To determine whether there was an impact of rs1049353 and rs2023239 genotype on blood THC concentrations, we first ran a genotype (major allele homozygous, minor allele carrier) by time (baseline, 5 min, 15 min, 30 min, 1 h, 2 h, 3 h, 4 h, 5 h, 6 h) ANOVA. For any significant genotype by time interaction or significant main effect of genotype, independent-samples *t*-tests were used to determine at which time points the genotypes significantly differed. Next, we ran *t*-tests to determine whether there was a significant genotype difference in THC area under the curve (AUC) or maximum concentration (Cmax). Greenhouse–Geisser corrections are reported whenever the sphericity assumption was violated for any analysis with a within-subjects term.

The next statistical analysis performed on the dataset was a comparison of maximum VAS and POMS scores between homozygotes for the major alleles and carriers of the minor alleles at the rs1049353 and rs2023239 SNPs. The VAS component consisted of 7 items, completed by participants at baseline, 5, 15, 30 min and 1, 2, 3, 4, 5, 6, 24, and 48 h post-dose. The POMS component consisted of 10 subscales, completed by participants at baseline, 1, 24, and 48 h post-dose. With a focus on studying the acute subjective effects of smoked cannabis, the 24 and 48 h time points were excluded from analysis. Therefore, using IBM SPSS MAX value function, maximum values for VAS items were determined for each participant from timepoints between baseline and 6 h post-smoking, and maximum values for POMS subscales were determined from timepoints at baseline and 1 h post-smoking.

To analyze the variance in maximum values between genotypes at both polymorphic sites, a univariate ANOVA model was utilized, in which the maximum VAS and POMS scores, as calculated for each subject, were the outcomes and genotype was the categorical predictor in the model. For each item, the ANOVA model numerically assessed genotypic significance and produced an estimated marginal means table displaying mean maximum values for each genetic polymorphism. Using the estimated marginal means and standard error calculated by SPSS, bar charts were constructed to display genotype group differences in maximum values of the VAS items and POMS subscales.

The next statistical analysis performed was a comparison of area under the curve (AUC) calculations of VAS results between major-allele homozygous and minor-allele-carrying genotype groups at the rs1049353 and rs2023239 SNPs. To calculate the AUC for each item, time course graphs were constructed using the same timepoints as for the maximum value analyses (baseline, 5, 15, 30 min and 1, 2, 3, 4, 5, and 6 h timepoints for VAS items). Similar to the maximum value analysis, the 24 and 48 h timepoints were excluded from the analysis.

A mixed-effects model (MEM) was used to construct time course graphs, where the test for difference in AUC was conducted using contrast of estimated marginal means of the time by genotype interaction. For each item, the MEM calculated the difference in AUC between major and minor polymorphic allele groups, a genotype interaction significance level, and an estimated marginal means table for each genotype comparison group. Using the estimated marginal means and standard error calculated by SPSS, time course graphs were constructed to display allelic differences in values of the VAS items over time.

The next statistical analysis involved constructing a split-plot ANOVA to analyze the magnitude and significance of POMS score movement between baseline and 1 h post-cannabis administration between genotypic groups. Within IBM SPSS, a split-plot ANOVA was constructed between these two timepoints and the genotype by time interaction was quantified.

Following analysis of the rs1049353 and rs2023239 SNPs with subjective data, adjusted models for each item were produced by controlling for variables deemed significant in relation to their effect on the subjective effects of smoked cannabis. Therefore, maximum value univariate ANOVA models, AUC mixed-effects, and split-plot ANOVA model results were adjusted by controlling for participant sex, BMI, and estimated THC dose administered during the CADRI trial.

Our primary hypothesis included two main outcomes related to rs1049353 VAS scores—maximum VAS score and Area Under the VAS Score Curve. Following analysis of the rs1049353 VAS scores, a Bonferroni correction for two tests was used, so in order to keep significance levels equal to 0.05, we declared significance if *p*-values were lower than 0.025. Our secondary hypothesis encompassed VAS scores for rs2023239 as well as POMS scores for both polymorphisms, and therefore these results did not require Bonferroni correction.

## 4. Discussion

Following analysis by *CNR1* rs1049353 genotype, our results display elevated subjective response in T-allele-carrying individuals (TC + TT) compared to CC individuals, demonstrated by greater VAS and POMS maximum scores, AUC values, and change from baseline. Despite seemingly elevated subjective readings for the T-allele group, only a subset of subjective effect items exhibited significant differences between rs1049353 genotypes. Of our maximum value results, none of the VAS items were significantly elevated in the T-allele group following adjustment for sex, BMI, and estimated THC dose administered. However, the *Anger*/*Hostility*, *Depression*, and *Vigor* POMS subscales retained significantly elevated results post-adjustment. Similarly, the AUC analysis of rs1049353 exhibits no VAS items displaying significance by genotype post-adjustment and correction. Similarly, split-plot ANOVA analysis to analyze the changes between baseline and post-cannabis between genotypes exhibited no significant results. These results suggest that the rs1049353 minor T-allele augments the subjective effects of smoked cannabis. While possible, it was also determined that rs1049353 T-allele carriers consumed more cannabis during the course of the 10 min smoking period, meaning this relationship needs to be further explored.

The rs1049353 *CNR1* polymorphism has been comprehensively studied, yet prior literature shows discrepant results regarding its association with cannabis use and subjective effects. Indeed, the T-allele has been associated with greater levels of cannabis use and multiple T-alleles further amplify cannabis use [23]. Cannabis users homozygous for the major C-allele have displayed greater satiety from cannabis, expressed as decreased desire for additional cannabis following THC exposure. T carriers do not experience these changes following cannabis administration, indicating a possible predisposition to develop cannabis use disorder [24]. Initial subjective reward following substance use has been studied and used to predict further usage patterns. Adolescents that recount a first cannabis experience characterized by predominantly positive subjective effects are 28.7 times more likely to develop lifetime cannabis dependence compared to adolescents recounting a neutral or negative first experience [38]. Alternatively, it may be inferred that T carriers experience weakened effects from cannabis usage, thereby experiencing less satiety following cannabis usage and desire further exposure. Considering the rs1049353 polymorphism is a silent mutation of the *CNR1* gene, it is unclear which of these is a plausible explanation.

In contrast, other research groups have discovered greater abundance of the CC rs1049353 genotype in cannabis users versus nonusers, and that the C-allele significantly increases the likelihood of cannabis use [25]. The C-allele has also been inconclusively associated with cannabis use disorder [27]. One investigation into the influence of rs1049353 on subjective effects of smoked cannabis utilized the POMS scale, and individuals with the homozygous CC genotype experienced greater *Anger*/*Hostility* scores following cannabis exposure, disagreeing with our results that clearly display elevated *Anger*/*Hostility* maximum and AUC values in T-allele-carrying individuals [26]. With the literature divided and our study suggesting minor T-allele-carrying individuals experience greater subjective effects, further investigation is needed.

The rs1049353 polymorphism is also involved in abuse of other substances. The T-allele has been associated with problematic drinking habits and alcohol use disorder, while genetic analysis found an association between the CC genotype and heroin dependence [39,40,41]. Ultimately, the endogenous cannabinoid system is integrally linked with reward pathways in the brain, and influence of CNR1 polymorphisms over the subjective effects and rewarding feelings from substance use seems logical. Although there are divided findings in the literature, our findings suggest that the rs1049353 T-allele enhances subjective effects of smoked cannabis, possibly linking it to substance use in general. Similar to how a positive initial experience with cannabis may predict future dependence, other substances have been studied in this capacity. Regarding alcohol consumption, decreased sensitivity to the effects of alcohol has been associated with possible future heavy drinking and alcohol-related problems due to increased consumption to achieve desired results [42,43]. Future tobacco dependence may be predicted by feelings of relaxation associated with first time cigarette smoking, and initial consumption of menthol cigarettes has been associated with a more positive first experience, possibly leading to further experimentation [44,45].

Another aspect to consider is the effect of the rs1049353 polymorphism on behaviour, as the minor T-allele has been associated with amplified impulsivity in various populations [46,47]. Reward and pleasure in response to substance use has been associated with heightened impulsive behaviour and increased substance use [48,49]. The genetic predisposition of T-allele carriers towards impulsive behaviour may help explain our overall trend towards elevated subjective response in this group. Contrarily, CC homozygous individuals are more likely to act impulsively and place larger bets during laboratory gambling tasks than T carriers. In spite of this finding, it is difficult to replicate impulsive behaviour in laboratory settings [50]. Impulsivity was not assessed in our study but may be worth assessing in future investigations.

Similar to the rs1049353 SNP, we found an association of the *CNR1* rs2023239 polymorphism with greater subjective effects in the minor allele CT + CC group compared to the TT group. Unlike rs1049353 that exhibits seemingly elevated results in the minor allele group, a small subset of VAS items and POMS subscales actually display elevated results in the rs2023239 TT group, albeit, none of these associations were significant. Of the maximum value results, no VAS subscales were significantly different pre- or post-adjustment, and none of POMS item were significantly different by genotype. The AUC analysis produced two VAS items *I like the drug* and *It feels like cannabis* that presented significantly larger AUC values in C-allele-carrying subjects post-adjustment. Similarly, split-plot ANOVA analysis to analyze the changes between baseline and post-cannabis between genotypes exhibited no significant results. From our results, there is some suggestion that the C-allele of the rs2023239 polymorphism generates elevated response to smoked cannabis. In spite of relative lack of significant genotypic differences, it is intriguing how, similarly for rs1049353, there is a nearly global elevation in minor allele subjective effect measure values.

The literature surrounding the rs2023239 polymorphism reveals minor C-allele-carrying cannabis users experience greater withdrawal after short-term cannabis abstinence, higher craving after smoking, greater activity in reward-related brain pathways following cannabis cue exposure, and increased cannabis usage [32,33]. This phenomenon may be explained by correlations of the C-allele with greater in vivo CB1 receptor density and ligand binding affinity that have been associated with the C-allele [29,31]. Augmented affinity and density of CB1 receptors could potentiate greater CB1 signaling from THC, therefore leading to elevated subjective effects from cannabis usage, a theory that is supported by our results. In spite of this finding, C-allele-carrying cannabis users have fewer cannabis-related problems, but only if they have low trait impulsivity [51]. Furthering this, the C-allele has been associated with greater subjective effects and craving intensity following alcohol consumption [30,52]. That being said, the minor allele has not been associated with substance abuse or dependence [53,54,55,56]. The literature strongly supports our findings that C-allele carriers trended towards greater subjective effects from smoked cannabis.

As with rs1049353, the POMS scale has been utilized to investigate the influence of rs2023239 allelic identity over the subjective effects of smoked cannabis [26]. The C-allele was associated with increased *Anger*/*Hostility* ratings following cannabis exposure. While we did not find significantly elevated *Anger*/*Hostility* ratings in C-allele carriers, there is a trend towards higher ratings in this genotypic group in both our maximum value and AUC analyses. In contrast to our results, Palmer et al. also reported lower *Confusion* readings in C carriers. Additionally, they reported decreased *Tension*/*Anxiety* and *Anger*/*Hostility* in T-allele carriers, which we observed. However, these results were not statistically significant. It is difficult to compare results as they utilized a regression analysis model, while we used the maximum value and AUC analyses. The rs2023239 polymorphism requires further investigation into its association with the subjective effects of cannabis.

One strength of this study is the vast amount of data generated. In terms of subjective effect data capture, there is a plethora of data, specifically 17 data points between the VAS items and POMS scales, all of which capture different aspects of subjective effects. Another strength of this study is that two separate statistical analyses were performed; maximum value and AUC, on each data point. Each of these analyses focused on different aspects of the data trajectory, one employing the absolute maximum reading, and the other considering the time course during acute cannabis intoxication. Performing both analyses strengthens our results, particularly if a subjective data item was significant following both analyses.

This study and its results are subject to important limitations. Firstly, the sample size of this study was small. From the original CADRI trial, there were 52 active cannabis group participants that consented to the secondary genetic analysis, and while this is not a particularly small sample size for a clinical trial, it is certainly small for a genotyping study [57,58,59,60,61]. Considering the limited contribution single-nucleotide polymorphisms (SNPs) have on their resultant protein products, genotyping studies generally require large subject pools to observe significant differences in genotypic groups. A larger sample size may have produced greater differences by genotype, as our results display many subjective items that trended towards significance. Additionally, it would allow for separation into three polymorphic groups: Major/Major, Major/minor, minor/minor, to assess possible dose-dependent allelic effects as other previous works have studied [23]. A larger sample size could also potentially allow for control of ancestry during statistical analysis, as it is well known they differ by allelic identity at numerous polymorphic sites [62,63,64]. Unfortunately, we were unable to control for ancestry in our analysis due to our restrictive sample size. The trends we observed in our analysis are encouraging and suggest a differing subjective response by genotype and these findings may be elucidated by larger sample size experimentation.

Other limitations that require discussion relate to the design of subjective data capture employed during the original study and study participant inclusion and exclusion criteria. Firstly, the VAS questions were completed by participants at numerous set timepoints post-cannabis exposure, while the POMS questionnaires were only completed 1 h post-exposure. The psychoactive effects from smoked cannabis peak between 15 and 30 min and may last up to 4 h, as such, subjective effect measurements should be assessed at timepoints within the first hour and extend to the end of the acute intoxication period [65,66]. The POMS scale is valuable in the assessment of substance-induced subjective effects, and therefore should ideally be completed at the same frequency as VAS questions. One caveat to this proposal is that experiments designed to monitor the acute phase of substance intoxication often operate within a relatively tight timeframe. For this reason, implementation of further questionnaires, particularly longer questionnaires like the POMS may prove difficult and impractical.

All participants enrolled in this study were regular cannabis smokers (1–4 times per week), an enrolment criterion that possibly influenced our results. Considering enrolled individuals were regular cannabis users, it suggests they may have an affinity for cannabis use, possibly due to a genetic predisposition. While we focused on allelic identity at the rs1049353 and rs2023239 polymorphic sites, by only studying regular cannabis users, we may have missed crucial differences that exist between regular users and those that are seldom or irregular cannabis users. Specifically, individuals that have experimented with cannabis yet are not regular users are not captured in our subject pool. While our experiment is a good first step, a grouping strategy that stratifies individuals by level of cannabis use, from sporadic, seldom use, to frequent daily use would be helpful to elucidate genetic differences. Furthermore, significant differences in cannabis tolerance may exist between individuals within our recruitment threshold of 1–4 cannabis usage sessions per week. Therefore, future analyses of allelic groups should aim to stratify by baseline smoking frequency, years of cannabis usage, and typical cannabis dosage per episode.

The most logical future direction to further this research would be to increase the sample size. This genotyping experiment was performed as a secondary study from a cannabis and driving study, and as such, the small sample size is reflective of the nature of the original clinical trial.

SNPs have relatively small influence on resultant protein products and the phenotypic characteristics they control. Recently published genotyping experiments studying SNPs in relation to reward processing and addictive behaviour have assembled samples ranging from 500 to nearly 25,000 subjects [57,58,59,60,61]. Future investigations into this topic should have a primary aim of studying the genetics behind the subjective effects of smoked cannabis, and as such, develop a sample size more appropriate for answering these experimental questions. Larger sample sizes will allow stratification of subjects into three genetic groups to analyze a potential additive effect of minor alleles and will allow for control of results by relevant parameters such as ancestry.

Finally, future investigations should attempt to vary the pool of participants such that various cannabis consumption patterns are represented. It would be worthwhile to analyze the genetics of infrequent cannabis smokers to elucidate a potential genetic basis regarding their lessened cannabis consumption compared to the frequent smokers enrolled in this study. It is possible that allelic identity of SNPs such as rs1049353 and rs2023239 may differ in frequency in cannabis users due to differing subjective effects and this relationship requires confirmation.

The purpose of this pilot study was to assess the allelic influence of the CNR1 rs1049353 and rs2023239 SNPs over the subjective effects of smoked cannabis. To study this relationship, we genotyped participants and utilized subjective data following cannabis exposure to perform a maximum value and area under the curve statistical analysis. Our results display a global trend towards greater subjective effects for rs1049353 T-allele- and rs2023239 C-allele-carrying subjects. However, significantly elevated readings were only documented for a subset of subjective effect items. These findings suggest that the minor alleles of both the rs1049353 and rs2023239 polymorphisms are associated with heightened subjective effects in response to smoked cannabis. With the literature divided on the effect of allelic identity at these polymorphic sites, this research is a meaningful contribution that requires replication at a larger scale.

## Figures and Tables

**Figure 1 ijms-22-07388-f001:**
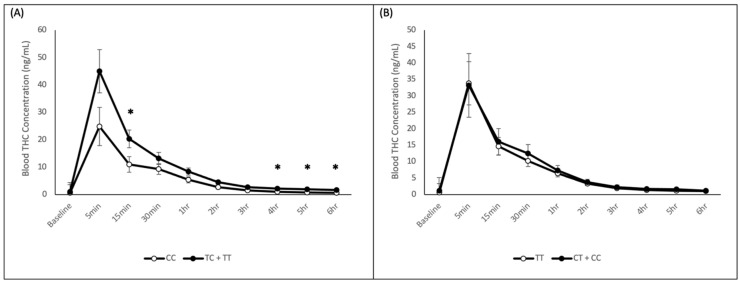
*CNR1* rs1049353 and rs2023239 Blood THC Concentration. Time course of mean blood THC concentrations of major allele and minor allele-carrying groups for the (**A**) rs1049353 and (**B**) rs2023239 polymorphism. * *p* < 0.05.

**Figure 2 ijms-22-07388-f002:**
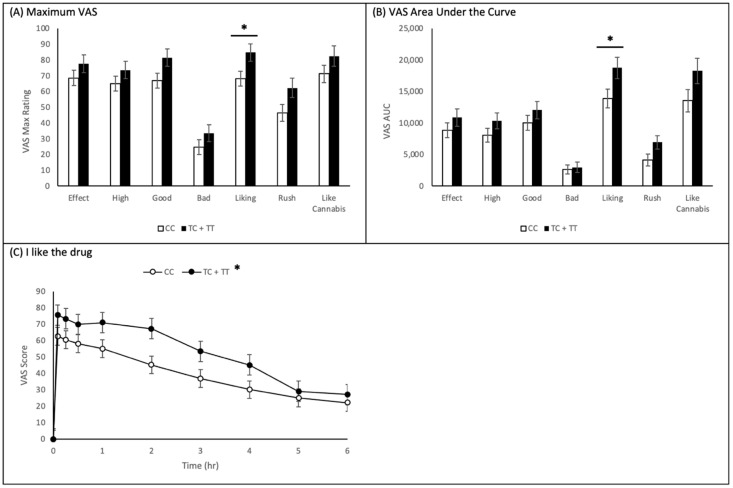
Impact of *CNR1* rs1049353 (TC + TT vs. CC) on Subjective Effects of Cannabis. VAS measurements (ranging from Not at all to Extremely (0 to 100)) were collected at different time points during acute cannabis intoxication (0–6 h) of *n* = 52 subjects, separated by rs1049353 genotype. (**A**) Maximum values of VAS scores obtained with various questions. Unadjusted maximum VAS question results: I feel a drug effect (*p* = 0.226), I feel this high (*p* = 0.235), I feel the drug’s good effects (*p* = 0.053), I feel the drug’s bad effects (*p* = 0.226), I like the drug (*p* = 0.027), I feel a rush (*p* = 0.057), and It feels like cannabis (*p* = 0.194). (**B**) Area under the curve analysis for VAS values collected. Unadjusted mixed-effects model AUC results: I feel a drug effect (*p* = 0.269), I feel this high (*p* = 0.181), I feel the drug’s good effects (*p* = 0.266), I feel the drug’s bad effects (*p* = 0.739), I like the drug (*p* = 0.036), I feel a rush (*p* = 0.054), and It feels like cannabis (*p* = 0.083). (**C**) Time course of the scoring for the VAS question ‘I like the drug’… Unadjusted mixed-effects model time course results of *I like the drug*. * *p* < 0.05, TC + TT vs. CC.

**Figure 3 ijms-22-07388-f003:**
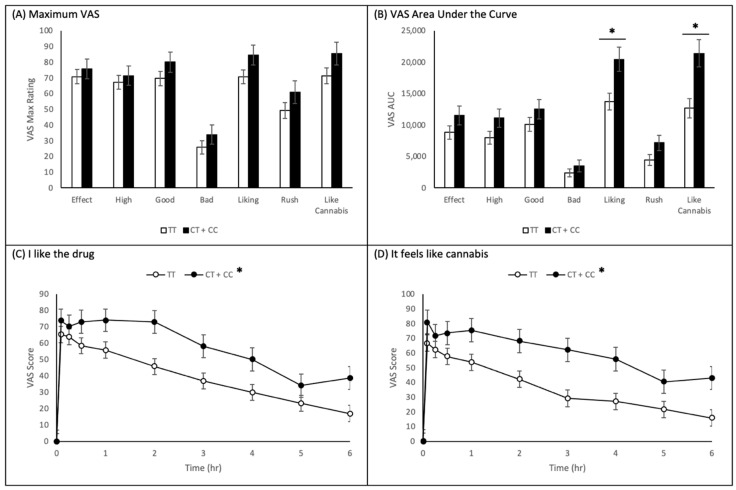
Impact of *CNR1* rs2023239 (CT + CC vs. TT.) on Subjective Effects of Cannabis VAS. Measurements (ranging from Not at all to Extremely (0 to 100)) were collected at different time points during acute cannabis intoxication (0–6 h) of *n* = 52 subjects, separated by rs2023239 genotype. (**A**) Maximum values of VAS scores obtained with various questions. Unadjusted maximum VAS question results: I feel a drug effect (*p* = 0.536), I feel this high (*p* = 0.591), I feel the drug’s good effects (*p* = 0.200), I feel the drug’s bad effects (*p* = 0.290), I like the drug (*p* = 0.080), I feel a rush (*p* = 0.193), and It feels like cannabis (*p* = 0.120). (**B**) Area under the curve analysis for VAS values collected. Unadjusted mixed-effects model AUC results: I feel a drug effect (*p* = 0.156), I feel this high (*p* = 0.082), I feel the drug’s good effects (*p* = 0.209), I feel the drug’s bad effects (*p* = 0.334), I like the drug (*p* = 0.006), I feel a rush (*p* = 0.071), and It feels like cannabis (*p* = 0.002). (**C**,**D**) Time course of the scoring for the VAS question (**C**) ‘I like the drug’ and (**D**) ‘It feels like cannabis’. Unadjusted mixed-effects model time course results of *I like the drug* and (**D**) *It feels like cannabis*. * *p* < 0.05, CT + CC vs. TT.

**Figure 4 ijms-22-07388-f004:**
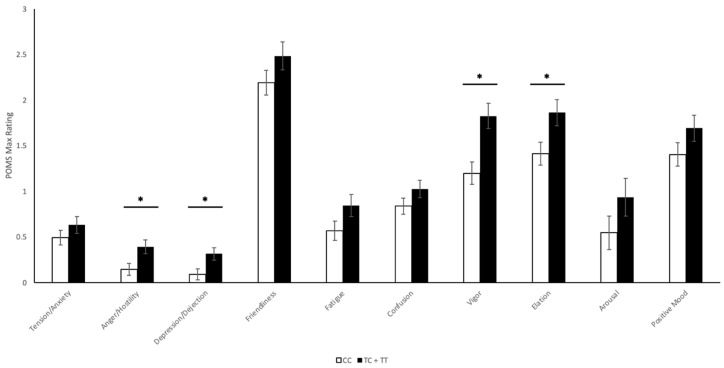
*CNR1* rs1049353 Aggregate Profile of Mood States Maximum Values. Unadjusted maximum Profile of Mood States (POMS) questionnaire results (measured on a scale of 0–4), measured at baseline and 1 h post-cannabis exposure of *n* = 52 subjects, separated by rs1049353 genotype: Tension/Anxiety (*p* = 0.261), Anger/Hostility (*p* = 0.018), Depression/Dejection (*p* = 0.016), Friendliness (*p* = 0.160), Fatigue (*p* = 0.087), Confusion (*p* = 0.155), Vigor (*p* = 0.001), Elation (*p* = 0.021), Arousal (*p* = 0.168), and Positive Mood (*p* = 0.146). * *p* < 0.05, TC + TT vs. CC.

**Figure 5 ijms-22-07388-f005:**
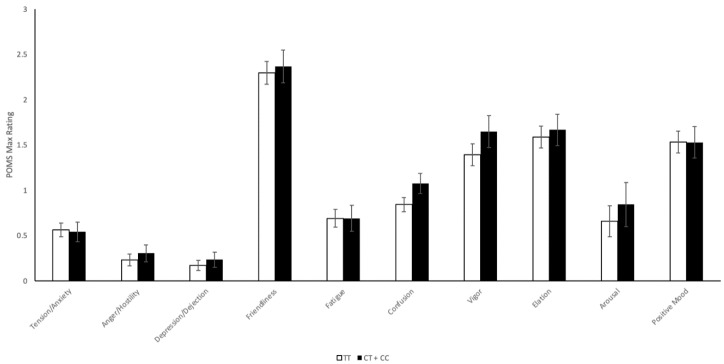
*CNR1* rs2023239 Aggregate Profile of Mood States Maximum Values. Unadjusted maximum Profile of Mood States (POMS) questionnaire results (measured on a scale of 0–4), measured at baseline and 1 h post-cannabis exposure of *n* = 52 subjects, separated by rs2023239 genotype: Tension/Anxiety (*p* = 0.876), Anger/Hostility (*p* = 0.511), Depression/Dejection (*p* = 0.526), Friendliness (*p* = 0.750), Fatigue (*p* = 0.999), Confusion (*p* = 0.098), Vigor (*p* = 0.239), Elation (*p* = 0.706), Arousal (*p* = 0.538), and Positive Mood (*p* = 0.988). CT + CC vs. TT.

**Table 1 ijms-22-07388-t001:** Participant demographic data.

Participant Demographics	Total (*n* = 52)	Male (*n* = 36)	Female (*n* = 16)
	Mean (SD)		
Age (years)	22.4 (1.9)	22.1 (2.0)	22.9 (1.5)
BMI (kg/m^2^)	24.6 (4.6)	25.2 (4.9)	23.1 (3.4)
Cannabis Smoking Frequency (times/week)	2.5 (0.9)	2.6 (0.8)	2.4 (1.1)

**Table 2 ijms-22-07388-t002:** Genotyping results and numerical representation of group size and composition following subject grouping by genotype at the CNR1 rs1049353 and rs2023239 SNPs.

*CNR1* Polymorphic Areas	Homozygous Individuals for the Major Allele		Carriers of the Minor Allele	
rs1049353	CC (*n* = 29)		TC + TT (*n* = 23)	
	Males (*n* = 19)	Females (*n* = 10)	Males (*n* = 17)	Females (*n* = 6)
rs2023239	TT (*n* = 35)		CT + CC (*n* = 17)	
	Males (*n* = 25)	Females (*n* = 10)	Males (*n* = 11)	Females (*n* = 6)

## Data Availability

The data presented in this study are available on request from the corresponding author. The data are not publicly available due to study participant confidentiality mandates.

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
