# Peer review of "Influence of Cannabinoid Receptor 1 Genetic Variants on the Subjective Effects of Smoked Cannabis"

_ijms, 2021, doi:10.3390/ijms22147388_

Round 1

Reviewer 1 Report

I reviewed the revised manuscript IJMS-1051594. The authors greatly improved this manuscript by adding important details to the materials and methods section, revising figures and figure legends, changing some analyses methods and removing extraneous analyses. I have a few very minor recommendations for edits before I believe the manuscript to be ready for publication:  

Original point 2. Information about recruitment (how participants were recruited, area).

Author’s response: Lines 170-172: The following statement was added, stating “Potential

participants were recruited within the Greater Toronto Area and initially

underwent a telephone eligibility screen with a member of the study team and

attended CAMH for an eligibility assessment.”

Reviewer’s response: Using fliers, ads, community events, other media, or any targeted areas (e.g., college campuses, clinics, etc)?

Original point 6. How was criteria for lifetime substance dependence (an exclusion criteria)

assessed?

Authors’ response: Within the materials and methods section it states lifetime substance

dependence was assessed using DSM-IV: “or met criteria for lifetime substance

dependence (DSM-IV)”

Reviewer’s response: Was this done using a SCID, or some sort of questionnaire, or how?

Reviewer’s new minor point: bullet points in the middle of the materials and methods section looks weird (in my opinion). I recommend rewriting in paragraph format.

Original point 2. Provide information describing the POMS scales (e.g., a brief description,

psychometrics)

Authors’ response: Line 195: The following statement was added – “Psychometric rating

scales used to assess a variety of transient mood states”

Reviewer’s response: Thank you for this description. I just noticed that the citation used is from a testing service. Can you add citations from peer-reviewed research literature of its validation and/or its use?

Original point: Point 9: Information about marijuana dosing is needed, as is an explanation about why

“estimated THC dose” is included as a variable in analyses. Why is it estimated and not known?

Shouldn’t it be the same for everyone in the study? And how was it estimated?

Author’s Response 9:

The THC dose (12.5% THC) is stated in Line 162 and again throughout the manuscript.

Participants followed a smoking procedure where they smoked as much as they desired during

a 10 minute period from a 750 mg cannabis cigarette. The remnants were then weighed and

the difference in weight was used to estimate the THC dose they consumed.

Reviewer’s response: To be clear, 12.5% THC is not the dose—it’s the concentration of THC in the marijuana. The information added upon revision about how the marijuana was administered is very helpful for understanding the dose and why the “estimated THC dose” needed to be included as a variable in the analyses. I’m failing to find information in the methods about the unsmoked remnants being weighed and subtracted from 750 mg to estimate dose. Can you please add this or tell me where it is?

Original Point 17: Table 1 reports cannabis smoking frequency for the entire sample and split by sex. It

would be greatly informative to show that smoking frequency did not differ among the groups

of interest (i.e., the allele groups). Also, was information on the amount of cannabis used per

episode collected, or the number of years each participant had used cannabis? If one type of

allele carrier tends to smoke larger doses each time they smoke, or has been smoking longer,

they may have developed more of a tolerance to the subjective effects that were evaluated in

the study, which could potentially confound the results of the study.

Author’s Response 17:

The original study population was composed of frequent cannabis users ie those using cannabis

1-4 times per week. Considering the population fell between this narrow range, we believe

displaying this information would prove redundant.

We did not collect other data about years smoked, dosage of cannabis, etc.

Reviewer’s response: There could be very large differences in exposure (and thus tolerance) between a person who smokes a small amount 1x a week and a person who smokes a large amount every other day. Such a difference could ultimately affect subjective effects of marijuana. For this reason, I disagree that 1-4 times per week is a narrow range. Comparison among alleles of smoking frequency and amount per episode of smoking should be provided. If that’s not possible, then the lack of such an analysis should be discussed as a limitation in the discussion section.

Original Point 19: In figure legends, please define what asterisks mean. In some time course graphs,

there is an asterisk over one time point. Is this the only time point at which there was a

significant effect?

Author’s Response 19:

Asterisks were utilized to display statistically significant differences by genotype. In the case of

the bar charts, the asterisks were placed over the bars that represent the minor polymorphic

allele carrying group if that group exhibited a statistically significant difference in effect scores.

Similarly, for the time course graphs, asterisks were placed over the graph representing the

minor polymorphic allele carrying group if there was a statistically significant difference from

the major allele group.

Reviewer’s response: So if I’m understanding correcting, the asterisk over time course graphs represents a significant main effect of genotype? An asterisk over a single time point doesn’t represent this well. Please remove it or possibly move it next to “CT+CC” to signify significant effect of genotype. Also provide in the figure legend an explanation of what the asterisk means, for example: *p<0.05, TT vs. CT+CC.

Original Point 20: Figure legends for POMS and ARCI state that these measures were taken 0-6 hours,

but materials and methods state they were taken only at baseline and 1 hour after dosing.

Authors’ Response 20:

The figure legends for POMS states that data collected during the acute cannabis intoxication

period (0-6 hours) are displayed, however the only POMS and ARCI readings that were taken

during this timeframe were at baseline and at 1 hour after dosing. We define the acute

intoxication period between baseline and 6 hours post-dosing, as we do for the VAS, and

display data points collected during that time. The wording in the captions has more to do with

how we define the acute intoxication period, and not that readings were taken hourly.

Reviewer Response: In each figure legend, please be more specific about exactly what’s displayed in the respective figure. Saying that the figure shows the max measurement during the acute intoxication period (0-6 hours) implies that 1) multiple measurements were taken over the course of 6 hrs after dosing, and/or 2) a measure was taken 6 hours after smoking, which can mislead the reader. Simply state in the figure legend that these data were taken 1 hour after intoxication.

Reviewer 2 Report

A revision of the article entitled “Influence of Cannabinoid Receptor 1 (CNR1) Genetic Variants on the Subjective Effects of Smoked Cannabis” was submitted.  The authors were very responsive to the concerns of the reviewers and the manuscript was extensively updated.  The manuscript is clear, the statistics are described in more detail and there is less confusion since the ARCI sections were deleted.  Only a few minor edits are needed.

The statistics for figure 1 are described in detail in the text but no indication of significance is provided on the graph in figure 1.

Author Response

This manuscript is a resubmission of an earlier submission. The following is a list of the peer review reports and author responses from that submission.

Round 1

Reviewer 1 Report

Abstract:

  • What is a regulated clinical trial?

Introduction:

  • Could be shortened, more to the point. As is, reads a bit like a review.
  • Page 2, line 88: “Genetic variation of the CNR1 gene may generate conformational changes of the resultant receptor…” Things the receptor interacts with, such as ligands, g-proteins, and beta-arrestins, generate conformation changes of cannabinoid receptors. A more accurate way to phrase the sentence would be “Genetic variation of the CNR1 gene may generate populational variations in the structure and function of the resultant receptor…”
  • Page 3, line 114: what is meant by “augmented”…CB1 receptor binding? Is this more binding, less binding?

Materials and Methods:

  • Pending journal’s requirements, I recommend putting materials and methods section after introduction, instead of just before references.
  • The materials and methods section is missing quite a lot of essential information:
    1. Statement that the study was approved by an IRB (not just the genotyping part).
    2. Information about recruitment (how participants were recruited, area).
    3. Source of marijuana.
    4. Describe the environment where marijuana was smoked and where assessments were conducted, particularly were they alone or with others?
    5. Some data was taken 48 hours after marijuana dosing…did participants remain on the premises the entire time?
    6. How was criteria for lifetime substance dependence (an exclusion criteria) assessed?
    7. More information is needed about the scales:
      1. VAS: minimum and maximum scale? (was it 0-100?)
      2. Provide information describing the POMS scales (e.g., a brief description, psychometrics)
  • I don’t know what the ARCI is, and I’m unable to infer what is or how it’s used or why it’s included in this study from the information provided. Does it assess how much the dose “feels like” each listed drug? What if someone has never used one of the listed drugs? How are they supposed to report how similar the cannabis dose feels to the drug? Please provide a brief description of the Inventory, an explanation for why it’s included, and psychometric information.
  1. More information is needed about how the genotyping was done (i.e., what is meant by “standard molecular biology techniques?” need to include source of materials used.)
  2. Information about marijuana dosing is needed, as is an explanation about why “estimated THC dose” is included as a variable in analyses. Why is it estimated and not known? Shouldn’t it be the same for everyone in the study? And how was it estimated?
  • In statistical analysis, there is only one post-dosing time point used for POMS and ARCI, so why is this consider a “maximum” value? It’s also a minimum value. And what is the point of determining and comparing AUC for curves that include only the baseline value and a single post-dose value? It unnecessarily complicates the data. It would make more sense to assess difference between groups at baseline and after smoking cannabis, and/or change from baseline.  
  • Page 19, line 612: how does univariate ANOVA produce a “genotype interaction significance level”? Interaction with what?
  • Sex differences in the effects of drugs of abuse are well accepted, but what is the rationale for adjustment for BMI?
  • I don’t understand why a Bonferroni correction was applied for only two tests for rs1049353 (VAS “max” score and AUC) when there is not a single outcome for these two tests—there are at total of 14. Even ignoring the POMS and ARCI data, it seems that a more stringent correction should be made.

Results:

  • IQ is reported in Table 1, but there’s no description of how IQ was assessed provided in the materials and methods section. It’s not clear why IQ is included.
  • Page 4, line 150: “participants were enrolled and randomized, of which…” should clarify that the participants were randomized to cannabis or placebo group.
  • Page 4, lines 151-152, states that 3 were excluded from the final analysis. Please provide an explanation for why they were excluded.
  • Table 1 reports cannabis smoking frequency for the entire sample and split by sex. It would be greatly informative to show that smoking frequency did not differ among the groups of interest (i.e., the allele groups). Also, was information on the amount of cannabis used per episode collected, or the number of years each participant had used cannabis? If one type of allele carrier tends to smoke larger doses each time they smoke, or has been smoking longer, they may have developed more of a tolerance to the subjective effects that were evaluated in the study, which could potentially confound the results of the study.
  • Data on race and ethnicity should be reported for the sample, and for the allelic groups to show that effects of alleles are not potentially confounded by race/ethnicity.
  • In figure legends, please define what asterisks mean. In some time course graphs, there is an asterisk over one time point. Is this the only time point at which there was a significant effect?
  • Figure legends for POMS and ARCI state that these measures were taken 0-6 hours, but materials and methods state they were taken only at baseline and 1 hour after dosing.
  • Figure legends for time course graphs are confusing because they state that these are these are results for Area Under the Curve calculations. I understand that AUC was determined from these curves, but the legend should clarify that these are time course data from which AUC calculations were conducted.
  • Can the names of the items or scales be added to the graphs themselves (maybe the titles of the graphs?) so that one doesn’t have to constantly refer to the figure legend to see what the graph is describing?
  • As mentioned about materials and methods, doing AUC for Figures 5, 6, 11, and 12 doesn’t make a lot of sense. These figures really don’t add much more than the “maximum” values figures, other than providing data on baselines and changes from baseline to 1 hour post-dosing.
  • Figure 5G (Vigor): Difference between allelic groups appears to be an effect of genotype and doesn’t appear to have much to do with cannabis (i.e., no change in either group caused by cannabis).

Discussion:

  • Page 15, Line 375-377: “Cannabis users homozygous for the major C-allele have displayed greater 373 satiety from cannabis, expressed as decreased desire for additional cannabis following THC 374 exposure. T-carriers do not experience these changes following cannabis administration, indicating a 375 possible predisposition to develop cannabis use disorder [24]. This link to cannabis use disorder is 376 intriguing as it implies greater subjective effects in T-carriers, and therefore greater craving for 377 cannabis, a theory in line with our results.”

Although it seems obvious that lower satiety à greater craving à more use, it’s not clear to me that lower satiety à greater subjective effects à more craving à more use. Furthermore, the results of the present study don’t support the role of greater subjective effects in this chain because the subjective effects that were enhanced in the T-carriers following cannabis use in this study were aversive (tension/anxiety, anger/hostility, and depression (I contend that “vigor” is an effect of genotype and not cannabis use)). The discussion section provides interesting information about previously found associations between genotype at this SNP and drug use and behaviors, but doesn’t really tie in with it their interesting findings that the T-carriers show increases in aversive moods after using marijuana. 

  • Page 15, line 413: “…place larger bets during laboratory gambling tasks…however it is difficult to replicate impulsive behavior in laboratory settings.” Doesn’t the first part of that sentence contradict the second?
  • Page 16, Line 419: “rs1049353…exhibits globally elevated results in the minor allele group…” this is expressed several times throughout the manuscript, and I think this is not an accurate interpretation of the data. Claims of “elevated” results in one group or another should not be made unless this clearly not due to chance (i.e., survives statistical analysis).

Reviewer 2 Report

The manuscript entitled “Influence of Cannabinoid Receptor 1 (CNR1) Genetic Variants on the Subjective Effects of Smoked Cannabis” describes the frequency of two alleles in CNR1 and their association with VAS, POMS and ARCI subjective measures.  The strength of the study is that two SNPs were evaluated and associated with the various subjective questionnaires about the psychoactive effects and the euphoric high of THC.  Although it was a relatively small sample size the data are important because it provides important, and difficult to obtain data since it requires coordination of THC/cannabis administration, blood draws for genotyping and real time assessment following exposure. 

Could the ARCI scales be described briefly in the Materials and Methods?  For the unfamiliar reader the VAS and POMS scale endpoints are obvious but not for ARCI. All that is needed is a one sentence description or clarification in the figure legend that subjects are responding to questions related to drug effects produced by the various classes.  It might also be good to include a small section in the discussion about what it means that Amphetamine ARCI subscale was significantly elevated in the T allele of CNR1 rs1049353.  As a comparison, it is clear what is meant by “tension/anxiety” in the POMS scale, but less clear what is meant by  “amphetamine” in the ARCI scale.

Use of the term “In contrast to previous results…” in lines 315-316 is confusing because there are so many previous results; please clarify to which results the present ones are in contrast.

Minor:

Some of the asterisks fall outside the boxes, making them difficult to see.